# *Astragalus membranaceus* Alters Rumen Bacteria to Enhance Fiber Digestion, Improves Antioxidant Capacity and Immunity Indices of Small Intestinal Mucosa, and Enhances Liver Metabolites for Energy Synthesis in Tibetan Sheep

**DOI:** 10.3390/ani11113236

**Published:** 2021-11-12

**Authors:** Xianju Wang, Changsheng Hu, Luming Ding, Yiguo Tang, Haiyan Wei, Cuixia Jiang, Qi Yan, Quanmin Dong, Abraham Allan Degen

**Affiliations:** 1State Key Laboratory of Grassland Agro-Ecosystem, School of Life Sciences, Lanzhou University, Lanzhou 730000, China; xjwang17@lzu.edu.cn (X.W.); huzhsh19@lzu.edu.cn (C.H.); weihy17@lzu.edu.cn (H.W.); jiangcx18@lzu.edu.cn (C.J.); yanq2016@lzu.edu.cn (Q.Y.); 2Qinghai Provincial Key Laboratory of Adaptive Management on Alpine Grassland, Qinghai University, Xining 810016, China; qmdong@qhmky.com; 3Sichuan Kangbaqing Agro-Pastoral Technology Group Co. Ltd., Chengdu 610000, China; 4Desert Animal Adaptations and Husbandry, Wyler Department of Dryland Agriculture, Blaustein Institutes for Desert Research, Ben-Gurion University of the Negev, Beer Sheva 8410500, Israel; degen@bgu.ac.il

**Keywords:** *Astragalus membranaceus*, rumen bacteria, Tibetan sheep, immunity indices, antioxidant capacity

## Abstract

**Simple Summary:**

*Astragalus membranaceus* is a widely used traditional Chinese herb that has been used by humans for hundreds of years. The Qinghai-Tibetan plateau (QTP) is regarded as one of the remaining ‘Green’ places in the world. With the fast-developing intensive livestock production, sustainable and environmentally-friendly practices are required urgently on the QTP. In the current study, Tibetan sheep were supplemented with the root of *Astragalus membranaceus* (AMT) to reduce the use of chemical veterinary drugs and antibiotics, and to examine the effect on rumen bacteria, the antioxidant capacities and immunity indices of small intestinal mucosa and meat tissue, and the liver metabolome responses.

**Abstract:**

Natural, non-toxic feed additives can potentially replace chemical medications and antibiotics that are offered sheep to improve performance. In the present study, Tibetan sheep were supplemented with the root of *Astragalus membranaceus* (AMT), a traditional herb used widely in China. Twenty-four male Tibetan sheep (31 ± 1.4 kg; 9-month-old) were assigned randomly to one of four levels of supplementary AMT: 0 g/kg (A_0_), 20 g/kg (A_20_), 50 g/kg (A_50_) and 80 g/kg (A_80_) dry matter intake (DMI). The A_50_ and A_80_ groups increased the diversity of rumen bacteria on d 14 and the relative abundances of fiber decomposing bacteria. Supplementary AMT upregulated the metabolism of vitamins, nucleotides, amino acids and glycan, and downregulated the metabolism of lipids and carbohydrates. In addition, supplementary AMT enriched rumen bacteria for drug resistance, and reduced bacteria incurring cell motility. In general, AMT supplementation increased the concentrations of catalase (CAT), superoxide dismutase (SOD) total antioxidant capacity (T-AOC) and secretory immunoglobulin A (sIgA) in the small intestinal mucosa and CAT and SOD in meat tissue. The liver tissue metabolome response showed that AMT in the A_80_ lambs compared to the A_0_ lambs upregulated the metabolites for energy synthesis. It was concluded that supplementary *A. membranaceus* increased the relative abundances of fiber decomposing bacteria and improved the antioxidant capacities and immunity indices of small intestinal mucosa and meat tissue in Tibetan sheep.

## 1. Introduction

Tibetan sheep (*Ovis aries*) are well adapted to the extremely harsh conditions of the alpine environment and, traditionally, are raised only on natural pasture. Today, as in the past, they play a key role in the livelihood of local pastoralists on the Qinghai-Tibetan Plateau (QTP) [1,2], as the mutton is an important local food and is popular in northwest China. Animal products from the QTP have gained the reputation of being ‘green and healthy’, because of the advertised clean grazing pastoral environment in which they are raised.

Antibiotics and many chemicals used to improve the performance of livestock have been banned in a number of countries. There is a long history in using herbs for medical purposes in China, some of which can possibly replace banned substances in livestock production. For example, *Astragalus membranaceus*, a key traditional herb, has been used for nearly 2000 years to enhance the immune system in humans [3]. Its root contains polysaccharides, which increased serum antibodies, cytokines and antioxidants in mice [4]. In addition, its leaves promoted intestinal proliferation of lactic acid bacteria and inhibited the growth of coliform bacteria in quail [5]. In a companion paper [6], supplementary *A. membranaceus* improved average daily gain, rumen fermentation, and serum antioxidant capacity and immunity indices of Tibetan sheep. However, there is no information on the effect of *A. membranaceus* on rumen bacteria composition, antioxidant capacities and immunity indices of small intestinal mucosa and meat tissue, and liver metabolome responses. The aim of this study was to fill this important gap.

## 2. Materials and Methods

### 2.1. Study Site and Preparation of Astragalus Membranaceus

The protocol and all procedures on the animals were approved by the Animal Ethics Committee of Lanzhou University, China (Protocol number: LZU20180818034). The study was done between October 20, and December 30, 2018, at the Haibei Demonstration Zone of Plateau Modern Ecological Animal Husbandry Science and Technology, Haibei, China (36°55′ N, 100°57′ E, 3170 m above sea level). The average daily air temperature was −4.4 °C during the study.

Dried roots of *A. membranaceus* (AMT) were purchased from a traditional Chinese herbal medicine trading market in Longxi County (Gansu Province, China), and were washed, dried, ground and passed through a 1 mm sieve. Five hundred grams of AMT powder were heated in distilled water (powder:water = 1:10) for 1 h, filtered through four layers of gauze, and the residue was heated again for 30 min (powder:water = 1:5). The two portions of liquid were mixed and lyophilized (Labconco FreeZone 7.5, Kingston, NY, USA) to determine polysaccharides, flavonoids and astragaloside (Table 1). Polysaccharides were detected by the phenol-sulfuric method. Briefly, 2 g of lyophilized powder were diluted in 100 mL distilled water and 200 mL 80% ethanol were added. The mixture was centrifuged at 2860× *g* for 20 min and the precipitate was dissolved in 100 mL of boiling water. Two mL of the solution were added to 1 mL 5% phenol and 5 mL sulfuric acid, shaken well for 5 min, allowed to stand for 10 min, and then heated in boiling water for 20 min. Polysaccharides content were determined colorimetrically at 485 nm (Molecular Devices SpectraMax M5, Thermo Fisher Scientifc, Waltham, MA, USA), with glucose as a standard. One gram of lyophilized powder was dissolved in 50 mL methanol and refluxed for 4 h. After cooling to room temperature, the solvent was shaken well and filtered. Twenty-five mL filtrate were dried and dissolved in 5 mL methanol to determine flavonoids concentration by high-performance liquid chromatography (HPLC, Column: XDB-C18, 5 µm, 4.6 × 150 mm; 1200 Series, Agilent Technologies, Santa Clara, CA, USA), with isoflavone glucoside as a standard. Two grams of lyophilized powder were diluted in 40 mL methanol, soaked overnight, and then refluxed under heat for 4 h with a cable extractor. The extracted solution was dried, and the residue was dissolved in 10 mL distilled water. Then *n*-butanol solution (water saturated) was added four times to extract astragaloside and the extract washed twice with ammonia water. The solvent was concentrated and dried, and the residue was dissolved in 5 mL distilled water to determine astragaloside on a high-performance liquid chromatography (HPLC) system (Column: WandaS1I C18 Superb, 5 µm, 4.6 × 150 mm; Agilent Technologies 1226 Infinity Quatpump) equipped with an evaporative light scattering detector (3300 ELSD, AIItech, Radnor, PA, USA), and with astragaloside IV as a standard.

### 2.2. Experimental Design

The design of this study was described earlier [6]. In brief, 24, 9-month-old male Tibetan sheep (31 ± 1.4 kg) were used in a completely randomized design. Each sheep was penned individually (each pen = 2.5 × 3.5 m) and was offered 225 g concentrate DM (Menyuan Yongxing Ecological Agriculture and Animal Husbandry Development Co. Ltd., Haibei, China) and ad libitum oat hay two times a day, at 07:00 and 17:00. After a 2-week adaptation period, the sheep were assigned randomly to one of four treatments (*n* = 6 per treatment) that differed in the level of supplementary AMT: control—0 g/kg (A_0_), 20 g/kg (A_20_), 50 g/kg (A_50_) and 80 g/kg (A_80_) dry matter intake (DMI) for 56 days. The AMT, mixed with 10 g concentrate, was fed separately to each sheep at 07:00. The amount fed each sheep was based on DMI from pre-trial intakes and was included in the total DMI. Water was available freely. The composition of the feed and AMT are presented in Table 1.

### 2.3. Data Collection and Sampling

Samples of concentrate feed, oat hay and *Astragalus membranaceus* root were collected every 2 weeks, dried at 105 °C for 48 h, ground to 1 mm size and mixed thoroughly to determine crude protein by Kjeldahl method (AOAC, 1990) [7], and neutral detergent fiber (NDF) and acid detergent fiber (ADF) by fiber analyzer (ANKOM A2000i, New York NY, USA) [8]. The NDF (exclusion of α-amylase and sodium sulfite) and ADF values included residual ash.

A rumen content sample was collected from each sheep on d 14 and d 56 before morning feeding using an oral-rumen tube (Anscitech, Wuhan, China). Approximately 80 mL were collected from each sheep of which the first 30 mL were discarded to minimize contamination from saliva. The rumen fluid was strained through four layers of cheesecloth, and stored in 10 mL tubes at −80 °C.

The sheep were slaughtered on d 57 prior to morning feeding by exsanguination following captive bolt stunning. Carcasses were weighed, chilled overnight and then approximately 100 g *longissimus thoracis et lumborum* muscle (LTL) and 50 g liver tissue were removed and stored at −80 °C. A strip in the middle of the small intestine of approximately 10 cm was removed and washed with saline solution. The mucosa was scraped off using a glass slide, put in a 1.5 mL tube (Eppendorf, GCS, New York, NY, USA) and stored at −80 °C.

### 2.4. Rumen Microbial DNA Extraction, Sequencing and Analysis

DNA extraction, polymerase chain reaction (PCR) amplification and sequencing data analysis followed Wei et al. (2021) [9]. One mL of thawed rumen fluid was used for DNA extraction using TIANamp DNA Kit (Tiangen Biotech Co., Ltd., Beijing, China). The DNA quality and quantity were verified by a DNA spectrophotometer (ND-2000, Nano Drop, Wilmington, DE, USA). The collected bacterial DNA was amplified by PCR, which was used as a template. The primers were designed according to the conserved region, and the end of the primers was added with a sequencing connector. The products were purified, quantified and homogenized to form a sequencing library. The targets in the V3–V4 region of the bacterial 16S rRNA gene were amplified using 338F (5′-ACTCCTACGGGAGGCAGCA-3′) and 806R (5′-GGACTACHVGGGTWTCTAAT-3′). Phusion High-Fidelity PCR Master Mix and GC buffer were used for PCR. The PCR amplification procedure was 95 °C for 5 min, 15 cycles (95 °C for 60 s, 50 °C for 60 s, 72 °C for 60 s) and 72 °C for 7 min and the PCR products were sequenced using an Illumina HiSeq platform to generate paired 250 bp reads. The analysis used BMKCloud (www.biocloud.net; accessed on 25 July 2021). Microbial diversity was based on Illumina HiSeq 2500 sequencing platform, using the paired-end (PE) sequencing method to construct a small fragment library. Operational taxonomic units (OTUs) for each sample were obtained at the 97% similarity level by QIIME (version 1.8.0) and was annotated by comparing with the Silva (bacteria) classification database. The rumen bacterial richness index (Chao1) and diversity indices (Shannon and Simpson) were determined using Mothur software (version v.1.30). Principal coordinates analysis (PCoA) was done to compare the similarity of species diversity among different samples by QIIME (version 1.8.0). Functional gene enrichment analysis, based on the Kyoto Encyclopedia of Gene and Genomes (KEGG) analysis, was done via PICRUSt software.

### 2.5. Meat and Intestinal Mucosa Analysis

Concentrations of superoxide dismutase (SOD), total antioxidant capacity (T-AOC), malonic dialdehyde (MAD) and catalase (CAT) of LTL muscle tissue and small intestinal mucosa were measured using commercial colorimetric assay kits (Bejing Sinouk Institute of Biological Technology, Beijing, China). The secretory immunoglobulin A (sIgA) of intestinal mucosa was determined by an enzyme-linked immunosorbent assay (ELISA) kit (Bejing Sinouk Institute of Biological Technology).

### 2.6. Liver Metabolome Analysis

Metabolome analysis on liver tissue in A_0_ and A_80_ groups used the UHPLC-QE-MS method (Biotree Biomedical Technology Co., LTD, Shanghai, China). Briefly, 25 mg of liver tissue were placed in an EP tube, and 500 μL extract solution (acetonitrile:methanol:water = 2:2:1, with isotopically-labelled internal standard mixture) were added. After vortexing for 30 s, the samples were homogenized at 35 Hz for 4 min and sonicated in an ice-water bath for 5 min. The homogenization and sonication cycles were repeated 2 times and then the samples were incubated for 1 h at −40 °C and centrifuged at 2000× *g* for 15 min at 4 °C. The resulting supernatant was transferred to a fresh glass vial for analysis. The quality control (QC) sample was prepared by mixing an equal aliquot of the supernatants from all of samples. LC-MS/MS analyses were performed using an UHPLC system (Vanquish, Thermo Fisher Scientific, Waltham, MA, USA) with a UPLC BEH Amide column (2.1 mm × 100 mm, 1.7 μm) coupled to Q Exactive HFX mass spectrometer (Orbitrap MS, Thermo Fisher Scientific). The mobile phase consisted of 25 mmol/L ammonium acetate and 25 mmol/L ammonia hydroxide in water (pH = 9.75) (A) and acetonitrile (B). The auto-sampler temperature was 4 °C, and the injection volume was 3 μL. The QE HFX mass spectrometer was used for its ability to acquire MS/MS spectra on information-dependent acquisition (IDA) mode in the control of the acquisition software (Xcalibur, Thermo Fisher Scientific). In this mode, the software continuously evaluates the full scan MS spectrum. The ESI source conditions were set as following: sheath gas flow rate as 30 Arb, Aux gas flow rate as 25 Arb, capillary temperature 350 °C, full MS resolution as 60,000, MS/MS resolution as 7500, collision energy as 10/30/60 in NCE mode, spray Voltage as 3.6 kV (positive) or −3.2 kV (negative).

The raw data were converted to the mzXML format using ProteoWizard and processed with an in-house program, which was developed using R and based on XCMS for peak detection, extraction, alignment, and integration. Then, an in-house MS2 database (BiotreeDB) was applied in metabolite annotation. The cutoff for annotation was set at 0.3.

### 2.7. Statistical Analysis

Bacterial data were log transformed before analyses but are reported as their natural values. Data were analyzed using PROC MIXED of SAS (Version 9.2, SAS Institute, Cary, NC, USA) with period and AMT level as fixed effects, and sheep as a random effect:Y_ij_ = μ + A_i_ + P_j_ + C(1)
where Y_ij_ = dependent variable; μ = overall mean response; A_i_ = fixed effect of AMT level, i = A_0_, A_20_, A_50_, A_80_; P_j_ = fixed effect of period, j = 14, 56; C = random effect of Tibetan sheep. An interactive matrix algebra procedure of SAS was used to generate coefficients for unequally spaced contrasts. Then, orthogonal polynomial contrasts were used to examine whether the responses to different AMT levels were linear or quadratic. Data are presented as least square means, and statistical significance was accepted at *p* < 0.05. The Tukey-Kramer test was used to separate means where significance existed.

## 3. Results

### 3.1. Ruminal Bacteria

The observed operational taxonomic units (OTUs) were reduced (*p* < 0.001) in the A_50_ and A_80_ groups on d 14, and in the A_80_ group on d 56 (*p* = 0.016; Table 2). The Chao1 index showed the same trend as the OTUs, but without statistical significance on d 56. The Shannon index was lower (*p* = 0.003) in the A_50_ and A_80_ groups than the A_0_ and A_20_ groups on d 14, and the reverse was true for the Simpson index (*p* = 0.014). There were significant (*p* < 0.05) linear effects on observed OTUs, Chao1 index, Shannon index and the Simpson index on d 14. The four groups did not differ in either the Shannon or Simpson index on d 56.

Ten phyla of rumen bacteria were identified, with Firmicutes and Bacteroidetes comprising approximately 85% of the total (Table 3). The relative abundance of Firmicutes increased linearly (*p* = 0.001) with increasing AMT and was higher (*p* = 0.006) in the A_80_ group than the other three groups on d 14, and was higher in the A_50_ and A_80_ groups than the A_0_ and A_20_ groups on d 56. The relative abundance of Bacteroidetes decreased linearly (*p* = 0.010) with increasing AMT and was higher (*p* = 0.047) in the A_0_ and A_20_ groups than in the A_50_ and A_80_ groups on d 14 and in the A_0_ group than the other three groups on d 56 (*p* = 0.003). The relative abundance of Kiritimatiellaeota increased linearly (*p* = 0.011) with increasing AMT and was higher (*p* = 0.027) in the A_50_ and A_80_ groups than in the A_0_ and A_20_ groups on d 14 and was highest (*p* = 0.033) in the A_50_ group on d 56. There was no difference in relative abundance among groups in the other seven phyla on either d 14 or d 56.

There were 13 genera of rumen bacteria identified, with *Rikenenllaceae_RC9_gut_group*, *uncultured_bacterium_f_F082* and *Christensenellaceae_R-7_group* the dominant ones (Table 4). The highest relative abundance of *Rikenenllaceae_RC9_gut_group* occurred in the A_50_ group on d 56, which increased linearly (*p* = 0.024) with increasing AMT, and the highest relative abundance of *Christensenellaceae_R-7_group* occurred in the A_80_ group on d 14, which also increased linearly (*p* = 0.006) with increasing AMT. All groups with AMT had a lower (*p* = 0.003) relative abundance of rumen *Prevotella_*1 than the A_0_ group on d 56; the A_80_ group had the highest relative abundances of *uncultured_bacterium_f_Muribaculaceae* and *Selenomonas_*3 on d 14 (*p* = 0.014 and <0.001, respectively) and the A_0_ and A_20_ groups had higher (*p* < 0.001) relative abundances of *Quinella* than the A_5_ and A_80_ groups on d 56. *Prevotella_*1 decreased linearly (*p* < 0.001) with increasing AMT on d 56; whereas, *uncultured_bacterium_f_Muribaculaceae* increased linearly (*p* = 0.003) with increasing AMT on d 14. The A_50_ group had the highest relative abundances of *uncultured_bacterium_o_WCHB1-41* on d 14 (*p* = 0.006) and d 56 (*p* = 0.007) and the highest relative abundance of *Desulfovibrio* on d 14 (*p* = 0.003). The relative abundance of *Ruminococcus_*2 was higher (*p* = 0.008) in the A_20_ group than the other three groups on d 56, and there was no effect of AMT on the other genera of rumen bacteria.

The clusterings in the principal coordinates analysis (PCoA) demonstrated an evident effect of *A. membranaceus* (Figure 1). The bacterial communities were clustered together on sampling d 14 and d 56 and the A_0_ and A_2_ groups were clustered together, as were the A_50_ and A_80_ groups.

Based on the Kyoto Encyclopedia of Gene and Genomes (KEGG) analysis, the A_80_ supplement enriched the relative abundances of rumen bacteria gene families involved in the metabolism of co-factors and vitamins, nucleotides and amino acids, biosynthesis of glycan and secondary metabolites, translation, drug resistance, endocrine system, cell growth and death, digestive system, and replication and repair when compared with the A_0_ group (Figure 2). The genes involved in signal transduction, membrane transport, lipid metabolism, cell motility and carbohydrate and energy metabolism were enriched in the A_0_ group.

### 3.2. Antioxidant Capacities and Immunity Indices of Meat Tissue and Small Intestinal Mucosa

The effects of AMT on antioxidant capacities of meat tissue and small intestinal mucosa of the sheep are presented in Table 5. The concentration of CAT in meat tissue was higher in the A_50_ group than in the A_0_ and A_20_ groups (*p* = 0.031) and of SOD was higher (*p* = 0.033) in the A_50_ and A_80_ groups than in the A_0_ group. The concentration of T-AOC in meat tissue increased linearly (*p* = 0.014) with an increase in AMT, but concentrations of T-AOC and MDA did not differ among groups. In the small intestinal mucosa, the concentration of T-AOC was higher (*p* = 0.041) in the A_50_ group than in the A_0_ and A_20_ groups; and all dietary intakes with supplementary AMT had higher (*p* = 0.002) SOD concentrations than the A_0_ group. The A_50_ and A_80_ groups had higher (*p* = 0.001) concentrations of of sIgA in the small intestinal mucosa than the A_0_ and A_20_ groups, but the concentrations of CAT and MDA did not differ among groups.

### 3.3. Liver Metabolome Responses

The OPLS-DA model was employed to assess the difference between the groups with (A_80_) and without (A_0_) supplementary AMT on a metabolomic level (Figure 3). The score scatter plot was generated from the peak area of identified metabolites in the two groups. Separation between A_0_ and A_80_ groups were clearly evident after 56 days of supplementary AMT.

The responding biomarkers in the major detected pathways with supplementary AMT are presented in Figure 4. Nine significant metabolic pathways were detected, including metabolism of histidine, niacin, methionine, fatty acids, glutamic acid, pentose, glutathione, purine and pyrimidine. The concentrations of AMP (adenosine monophosphate), NAD^+^ (nicotinamide adenine dinucleotide) and NADP^+^ (nicotinamide adenine dinucleotide phosphate) increased in liver with supplementary AMT. A number of biomarkers decreased with AMT intake, including aspartate, L-aspartate, L-asparagine, L-methionine, α-linolenic acid, palmitic acid, arachidonic acid, D-xylose, hypoxanthine, deoxycytidine and thymidine.

The changes in metabolites after AMT supplementation are summarized in a heat map (Figure 5). Generally, supplementary AMT induced significant decreases in a number of metabolites, including acetylcysteine, dodecenoic acid, xylose, sedoheptulose, glutamylglutamic acid, deoxycytidine, bovinic acid, α-linolenic acid, hypogeic acid, pentadecanoic acid, pentadecanoic acid, pelargonic acid, caprylic acid, ethyl tetradecanoate, L-aspartic acid, palmitic acid, thymidine, xanthine, hypoxanthine, *trans*-urocanate, arachidonic acid, γ-glutamylglutamine, fexofenadine, propionylglycine, L-asparagine, L-methionine, myristoleic acid, capric acid, tridecanoic acid, and ethyl docanoate. Supplementary AMT increased the concentrations of levocetirizine, diethyl phthalic acid, paliperidone, NAD, AMP and NADP.

## 4. Discussion

### 4.1. Ruminal Bacteria

Natural plants and extracts can alter the gastrointestinal and rumen microbiome [10]. In turn, the rumen microbiota composition and activity can influence the performance, health, and immune system of the host [11]. The reduction of ruminal OTUs and alpha diversity, especially on d 14, with supplementary AMT at 50 and 80 g/kg DMI was mainly due to the flavonoids and saponins. Studies have demonstrated that plant extract flavonoids reduced the population of almost all rumen microorganisms by inhibiting cytoplasmic membrane function, cell wall synthesis and nucleic acid synthesis [12]. In vitro studies showed that dietary saponins influenced rumen fermentation and microbial composition [13]. The recovery on d 56 in the present study was most likely the adaptive effect of rumen microorganisms to supplementary AMT with time, as the PCoA analysis clearly demonstrated differences in rumen bacteria due to sampling times (d 14 vs. d 56).

Firmicutes are important fibrous decomposing bacteria that correlate positively with average daily gain, while Bacteroidetes are the major contributors of carbohydrate-active enzymes that promote the breakdown of structural polysaccharides (hemicellulose) in the rumen [10,14,15,16]. In the present study, supplementary AMT increased the relative abundance of rumen Firmicutes, and reduced the relative abundance of Bacteroidetes, which indicated that AMT promoted fiber digestion.

Few studies have reported on Kiritimatiellaeota, which was only recently recognized as a distinct phylum, as it was assigned previously to Verrucomicrobia. This phylum occupies predominantly the intestine of animals, but its function remains unknown [17]. In the present study, supplementary AMT at 50 and 80 g/kg DMI increased the relative abundance of Kiritimatiellaeota on d 14, and 50 g/kg DMI increased the relative abundance on d 56.

The main function of the genus *Rikenellaceae_RC9_gut_group* is in the fermentation of carbohydrates and proteins [18]; whereas the genus *Christensenellaceae_R-7_group* belongs to the rumen cellulolytic bacteria [19,20]. In a previous study, *Christensenellaceae_R-7_group* increased with the intake of saponins, and was correlated positively with metabolites such as citrulline, lanosterol, and squalene [21]. In the present study, supplementary AMT at 50 and 80 g/kg DMI increased the relative abundance of *Christensenellaceae_R-7_group* on d 14 and of *Rikenellaceae_RC9_gut_group* on d 56, which indicated the potential contribution of AMT to the degradation of carbohydrates, proteins, and fibers.

Supplementary AMT at 80 g/kg DMI increased the rumen relative abundances of the two genera, *uncultured_bacterium_f_Muribaculaceae* and *Selenomonas_3* on d 14. *Muribaculaceae* degrade polysaccharides, such as plant glycans and α-glucans, which undergo acetogenesis [22]; whereas *Selenomonas* is a starch-degrading and lactate-utilizing bacterium that reduces acid accumulation in the rumen [23,24]. *Ruminococcus_2* is one of the unnamed genera in the family of *Ruminococcaceae*, which is responsible for decomposing fibers [25]. *Desulfovibrio* belongs to the sulfate-reducing bacteria and can convert sulfate to hydrogen sulfide (H_2_S), which plays an important role in the mucosal defense of the digestive tract [26,27]. In the present study, supplementary AMT at 50 g/kg DMI increased the relative abundance of *Desulfovibrio* on d 14; consequently, the immunity promoting effect of AMT on the small intestinal mucosa may be associated with the relative abundance of *Desulfovibrio* and its metabolite H_2_S.

*Prevotella_1* belongs to the phylum Bacteroidetes, which degrades and utilizes plant cell wall polysaccharides, including hemicellulose, xylan and pectin in the rumen [28,29]. It was also reported that *Prevotella* has the capacity to utilize starch, simple sugars, and other non-cellulosic polysaccharides as energy sources [30]. Supplementary AMT reduced the relative abundance of *Prevotella_1* on d 56, probably due to the astragalus polysaccharide inhibiting its growth, which is consistent with the reduction in the relative abundance of Bacteroidetes. It was reported that fucoidan-rich extract reduced the relative abundance of *Prevotella* in the caecum of pigs [31] and that the reduction of *Prevotella* indicated a low incidence of sub-acute rumen acidosis in lactating cows [32]. In contrast, an increase in the relative abundance of *Prevotella* has been observed in the rumen of cattle facing acidotic challenge [33] and an increase in both *Prevotella* and *Quinella* has been reported in Holstein cows with severe ruminal acidosis [32]. *Quinella* is well adapted to acidotic conditions [34]; consequently, the reduction in *Quinella* with supplementary AMT at 50 and 80 g/kg DMI indicated a non-acidotic rumen condition.

The KEGG pathway revealed that vitamins, nucleotides, amino acids and glycan metabolism were upregulated, and that lipid and carbohydrate metabolism were downregulated by AMT supplementation at 80 g/kg DMI. These results are consistent with the decreasing relative abundance of the carbohydrate decomposing bacteria Bacteroidetes in the present study. Studies have shown that *Astragalus* polysaccharides influenced glucose, lipid, protein and nucleotide metabolism in cocks [35], and improved energy and protein metabolism, ameliorated amino acid metabolism and increased the entry of dietary amino acids into the systemic circulation in broilers and mice [36,37]. The present results showed that supplementary AMT enriched rumen bacteria for drug resistance, which was referred to as antimicrobial, due to flavonoids in AMT [12]. The enhancement of biosynthesis of glycan and secondary metabolites in the AMT groups was due to the content of polysaccharides and other functional secondary metabolites, such as flavonoids and saponins. However, the enhanced metabolism of co-factors, vitamins, nucleotides, and amino acids, translation, and replication and repair with supplementary AMT indicated high fluctuation and fast turnover rates [38]. This explained the lower rumen bacterial OTUs in A_80_ than in A_0_ group. The KEGG data showed that supplementary AMT inhibited rumen bacterial energy and carbohydrate and lipid metabolism compared with the control group. We reasoned that this effect could conserve feed energy for host utilization rather than for rumen bacteria.

### 4.2. Antioxidant Capacities of Meat Tissue and Small Intestinal Mucosa and Secretory Immunoglobulin A (sIgA) of Small Intestinal Mucosa

*Astragalus* polysaccharide in AMT has been reported to accelerate muscle growth through the expression of related genes and proteins [39]. Sheep produce ‘red meat’ due to its high myoglobin concentration, which predisposes meat to oxidation [40]. Consequently, enhancing antioxidant capacities of meat tissue by feeding supplementary herbs is an efficient and convenient management strategy [41]. Antioxidants scavenge oxygen-derived free radicals to protect tissues from injury [42]. The superoxide anion is first degraded into hydrogen peroxide by SOD and, subsequently, catalyzed and converted into water and oxygen by antioxidants such as CAT [43]. In the present study, AMT supplements of 50 and 80 g/kg DMI increased SOD concentrations in muscle, and of 20, 50 and 80 g/kg DMI increased SOD concentrations in small intestinal mucosa in sheep. In addition, AMT supplement of 50 g/kg DMI AMT increased the concentrations of muscle CAT and of small intestinal mucosa T-AOC. These results indicate that AMT can lessen oxidative reactions and protect muscle injury in sheep. We are unaware of other reports of the effect of AMT on antioxidant capacities of meat tissue and small intestinal mucosa. Zhong et al. [44] reported that 50 g AMT/kg DMI increased serum T-AOAC but did not affect CAT in lambs and Wang et al. [6] reported equivocal results of the effect of AMT on serum T-AOAC.

Secretory IgA (sIgA) has been recognized as the first line of defense against enteric pathogens and toxins in the lumen, preventing the growth and spread of intestinal commensal bacteria and the influenza virus, which cover the mucosal surface [45,46]. The present study demonstrated that an AMT supplement of 50 and 80 g/kg DMI enhanced small intestinal mucosa sIgA secretion, which we reasoned could be attributed to *Astragalus* polysaccharides. Studies have shown that mannan oligosaccharide protects the mucosa by stimulating intestinal cells to secrete immunoglobulins [47].

### 4.3. Liver Metabolome Response

The increased abundances of AMP, NAD and NADP with supplementary AMT indicated that AMT enhanced energy metabolism, because these metabolites are the intermediate products in the production of ATP (adenosine triphosphate). The inhibiting effect of AMT on energy metabolism of rumen bacteria could be related to the enhancing effect on liver energy metabolism. *Astralagus membranaceus* and its component formononetin increased adipocyte thermogenesis and energy expenditure of obese mice [48]. Components isolated from *A. membranaceus*, such as polysaccharides and astragaloside, improved energy metabolism and mitochondria function by reducing the accumulation of plasma free fatty acids and lactic acid [49,50,51,52]. In addition, AMT downregulated some liver saturated fatty acids, including tridecanoic acid, capric acid, myristoleic acid, palmitic acid, caprylic acid, pelargonic acid, pentadecanoic acid, and dodecenoic acid, and unsaturated fatty acids, such as arachidonic acid and α-linolenic acid. Fatty acids are a source of energy and, therefore, the current study indicated that the synthesis of energy stored components are mainly from the decomposition of body fatty acids. Chinese herbal medicines, including *A. membranaceus*, reduced plasma fatty acids and triglyceride synthesis in broilers [53] and free fatty acids content in rat heart tissue [54,55].

Aspartate and asparagine are involved in the citrate cycle, and in the metabolism of histidine, methionine and glutamic acid. The current study showed decreased concentrations of aspartate, L-aspartate and L-asparagine, which was accompanied by a decrease in L-methionine [56]. The decrease of aspartate promotes pyruvate carboxylase activity and produces more intermediates to the citrate cycle [57]. AMT downregulated pentose metabolism by decreasing the abundance of D-xylose, but the mechanism needs further research. The downregulation of xanthine and hypoxanthine by supplementary AMT suggests an inhibiting effect on the production of uric acid from the catabolism of xanthine and hypoxanthine [58]. The metabolites of deoxycytidine and thymidine in the pyrimidine metabolism pathway were identified as indicators of cardiotoxicity [59]. This finding could be linked with the previous stated cardiac protection by *A. membranaceus* [48,49].

## 5. Conclusions

Supplementary *Astralagus membranaceus* fed to Tibetan sheep increased the ruminal bacteria community by enhancing the abundance of fiber-degrading bacteria, improving antioxidant capacities and intestinal mucosa immunity indices of small intestinal mucosa and meat tissue, and promoting liver energy metabolism. Supplementary AMT at 50 to 80 g/kg DMI produced the most promising results.

## Figures and Tables

**Figure 1 animals-11-03236-f001:**
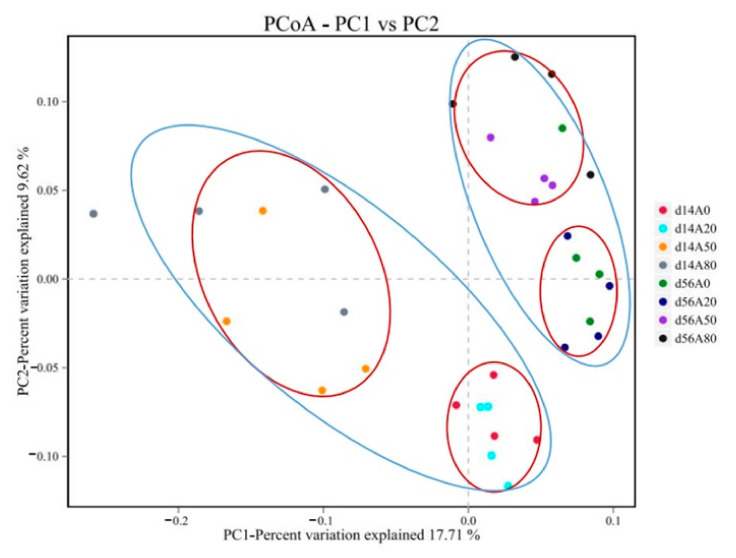
Rumen bacteria principal coordinates analysis (PCoA) of Tibetan sheep.

**Figure 2 animals-11-03236-f002:**
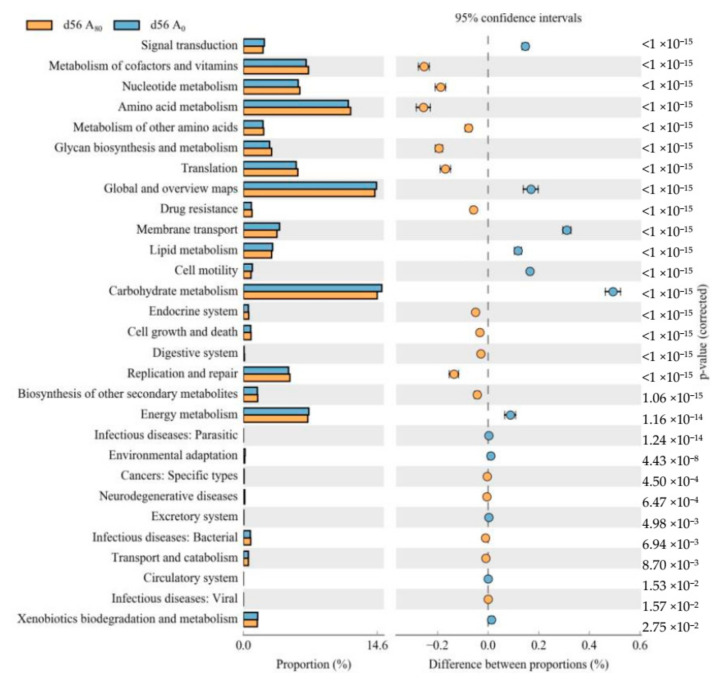
Kyoto Encyclopedia of Gene and Genomes (KEGG) analysis of rumen bacteria between group A_0_ and group A_80_ on d 56.

**Figure 3 animals-11-03236-f003:**
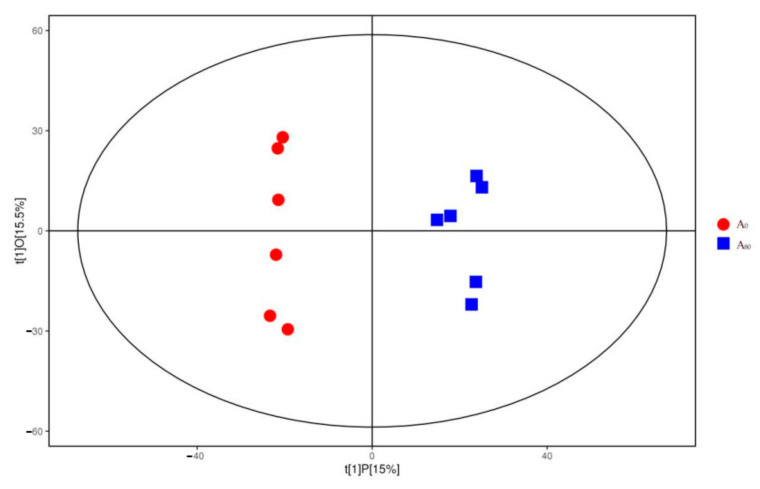
The score scatter plots acquired by the Orthogonal partial least squares discriminant analysis (OPLS-DA) model of the liver tissue of Tibetan sheep with (A_80_) or without (A_0_) supplementary *Astralagus membranaceus* (AMT).

**Figure 4 animals-11-03236-f004:**
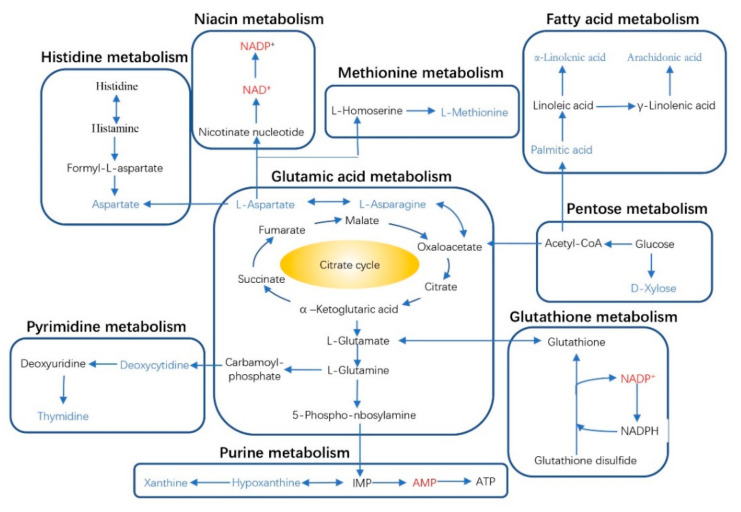
Schematic diagram of metabolic pathways with supplementary *Astralagus membranaceus* (AMT) (80 g/kg DMI, A_80_) in Tibetan sheep. Compared with control (without supplementary AMT), biomarkers with red font indicate an increase in the abundance of the responding biomarkers, blue fonts indicate a decrease in the abundance of the responding biomarkers, and black fonts represent no change.

**Figure 5 animals-11-03236-f005:**
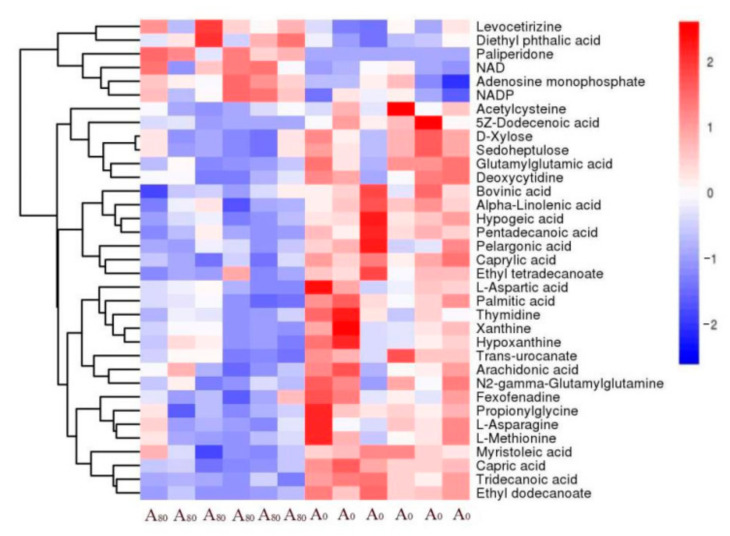
Heat map demonstrating the changes in concentration of metabolites with (A_80_) and without (A_0_) supplementary *Astralagus membranaceus* (AMT). The *y*-axis shows the 35 identified metabolites and the *x*-axis lists the AMT treatments. Red color squires represent high concentrations of metabolites, and blue color squires represent low concentrations of metabolites.

**Table 1 animals-11-03236-t001:** Composition (DM basis) of the concentrate, oat hay and *Astragalus membranaceus* root (AMT).

Ingredient (g/kg DM)	Concentrate ^1^	Oat Hay	AMT
Crude protein	201	84	194
Neutral detergent fiber	- ^2^	485	523
Acid detergent fiber	-	392	416
Polysaccharides	-	-	125.8
Flavonoids	-	-	0.12
Astragaloside	-	-	0.87

^1^ Concentrate included corn, wheat bran, soybean meal, canola meal, cottonseed meal, vegetable oil, NaCl, CaCO_3_, CaHPO_4_∙2H_2_O, amino acids and compound premix. ^2^ Not measured.

**Table 2 animals-11-03236-t002:** The effect of supplementary *Astralagus membranaceus* (AMT) on operational taxonomic unit (OTU) richness and alpha diversity of rumen bacteria of Tibetan sheep.

Items	Day	Treatment ^1^				SEM	*p*-Value ^2^		
		A_0_	A_20_	A_50_	A_80_		T	L	Q
OTUs	14	866 ^a^	890 ^a^	786 ^b^	802 ^b^	12	<0.001	0.004	0.012
	56	867 ^a^	885 ^a^	882 ^a^	826 ^b^	8	0.016	0.541	0.162
Chao1 index	14	951 ^a^	978 ^a^	889 ^b^	887 ^b^	46	0.003	0.003	0.010
	56	968	964	967	936	30	0.665	0.548	0.238
Shannon index	14	5.046 ^a^	5.319 ^a^	4.678 ^b^	4.688 ^b^	0.343	0.003	0.016	0.041
	56	4.998	5.152	4.989	4.900	0.227	0.391	0.878	0.307
Simpson index	14	0.019 ^b^	0.013 ^b^	0.031 ^a^	0.030 ^a^	0.010	0.014	0.019	0.063
	56	0.022	0.019	0.024	0.027	0.004	0.705	0.646	0.432

^a,b^ within a row followed by different lower-case letters indicate that the values differ significantly from each other (*p* < 0.05). ^1^ A_0_ contains 0 g AMT/kg DMI (DM basis); A_20_ contains 20 g AMT/kg DMI (DM basis); A_50_ contains 50 g AMT/kg DMI (DM basis); A_80_ contains 80 g AMT/kg (DM basis). ^2^ T = Treatment; L = Linear of effect of AMT; Q = Quadratic effect of AMT.

**Table 3 animals-11-03236-t003:** The effect of supplementary *Astralagus membranaceus* (AMT) on rumen bacteria composition at the phylum level of Tibetan sheep.

Items (%)	Day	Treatment ^1^				SEM	*p*-Value ^2^		
		A_0_	A_20_	A_50_	A_80_		T	L	Q
Firmicutes	14	39.79 ^b^	41.38 ^b^	48.90 ^b^	58.88 ^a^	2.427	0.006	0.001	0.002
	56	37.22 ^b^	39.07 ^b^	42.73 ^a^	46.28 ^a^	1.334	0.043	0.013	0.054
Bacteroidetes	14	46.48 ^a^	47.85 ^a^	32.97 ^b^	29.13 ^b^	3.192	0.047	0.010	0.039
	56	50.91 ^a^	46.18 ^b^	45.11 ^b^	36.59 ^c^	2.176	0.003	0.001	0.004
Patescibacteria	14	3.52	4.04	3.70	3.34	0.435	0.965	0.242	0.208
	56	3.37	4.17	4.17	4.81	0.375	0.666	0.053	0.142
Proteobacteria	14	3.19	2.46	3.50	2.46	0.421	0.812	0.775	0.949
	56	1.67	2.90	2.28	5.81	0.714	0.168	0.060	0.126
Kiritimatiellaeota	14	0.70 ^b^	1.34 ^b^	4.57 ^a^	3.80 ^a^	0.601	0.027	0.011	0.035
	56	1.01 ^b^	1.56 ^b^	2.62 ^a^	0.99 ^b^	0.262	0.033	0.686	0.093
Synergistetes	14	0.44	0.69	1.03	0.51	0.151	0.585	0.247	0.494
	56	2.17	2.09	1.26	1.09	0.435	0.801	0.211	0.421
Actinobacteria	14	0.92	1.07	1.80	0.82	0.185	0.242	0.251	0.363
	56	0.35	0.11	0.32	0.15	0.055	0.334	0.446	0.203
Tenericutes	14	1.22	0.64	0.93	0.96	0.225	0.882	0.818	0.805
	56	0.52	0.95	0.64	1.06	0.115	0.326	0.210	0.476
Cyanobacteria	14	0.34	0.18	1.46	0.27	0.229	0.149	0.630	0.512
	56	0.07	0.03	0.24	0.18	0.041	0.268	0.158	0.386
Spirochaetes	14	0.74	0.16	0.66	0.07	0.171	0.439	0.350	0.661
	56	0.15	0.20	0.44	0.11	0.075	0.467	0.299	0.178

^a–c^ within a row followed by different lower-case letters indicate that the values differ significantly from each other (*p* < 0.05). ^1^ A_0_ contains 0 g AMT/kg DMI (DM basis); A_20_ contains 20 g AMT/kg DMI (DM basis); A_50_ contains 50 g AMT/kg (DM basis); A_80_ contains 80 g AMT/kg (DM basis). ^2^ T = Treatment; L = Linear of effect of AMT; Q = Quadratic effect of AMT.

**Table 4 animals-11-03236-t004:** The effect of supplementary *Astralagus membranaceus* (AMT) on rumen bacteria composition at the genus level of Tibetan sheep.

Items (%)	Day	Treatment ^1^				SEM	*p-*Value ^2^		
		A_0_	A_20_	A_50_	A_80_		T	L	Q
*Rikenellaceae_RC9_gut_group*	14	15.19	15.00	15.59	15.39	0.573	0.989	0.823	0.976
	56	12.93 ^b^	13.09 ^b^	15.61 ^a^	14.36 ^b^	0.350	0.006	0.024	0.044
*uncultured_bacterium_f_F082*	14	10.43	12.80	11.92	13.39	0.823	0.653	0.292	0.566
	56	16.70	16.66	17.51	18.44	0.429	0.448	0.116	0.257
*Christensenellaceae_R-7_group*	14	10.64 ^b^	10.12 ^b^	12.78 ^b^	15.55 ^a^	0.763	0.026	0.006	0.010
	56	3.73	3.71	3.96	3.64	0.206	0.964	0.995	0.943
*Prevotella_1*	14	3.55	3.35	3.04	3.41	0.140	0.665	0.578	0.539
	56	7.67 ^a^	5.95 ^b^	4.64 ^b^	4.50 ^b^	0.402	0.003	<0.001	0.001
*uncultured_bacterium_f_ Muribaculaceae*	14	4.87 ^b^	5.02 ^b^	5.24 ^b^	8.28 ^a^	0.434	0.002	0.003	0.001
	56	3.74	4.03	3.18	3.04	0.238	0.604	0.002	0.001
*Ruminococcaceae_NK4A214_ group*	14	5.25	5.84	6.84	4.98	0.376	0.332	0.955	0.283
	56	2.06	3.29	3.87	2.63	0.293	0.133	0.400	0.063
*Succiniclasticum*	14	1.32	2.55	2.30	3.15	0.366	0.380	0.110	0.283
	56	5.33	5.10	6.37	5.06	0.376	0.615	0.896	0.790
*Quinella*	14	1.71	1.84	2.41	3.78	0.583	0.625	0.205	0.402
	56	3.40 ^a^	3.52 ^a^	1.95 ^b^	2.15 ^b^	0.208	<0.001	0.001	0.603
*Ruminococcus_2*	14	2.13	1.94	1.68	3.50	0.342	0.245	0.221	0.154
	56	1.01 ^b^	3.45 ^a^	2.08 ^b^	0.88 ^b^	0.339	0.008	0.582	0.011
*uncultured_bacterium_ o_WCHB1-41*	14	1.37 ^b^	1.87 ^b^	4.30 ^a^	1.88 ^b^	0.371	0.006	0.247	0.059
	56	0.93 ^b^	1.37 ^b^	2.62 ^a^	1.09 ^b^	0.216	0.007	0.395	0.037
*Veillonellaceae_UCG-001*	14	0.11	0.27	0.10	0.56	0.071	0.051	0.053	0.085
	56	1.96	2.52	2.69	2.90	0.209	0.463	0.111	0.269
*Desulfovibrio*	14	0.93 ^b^	1.17 ^b^	2.68 ^a^	1.78 ^b^	0.212	0.003	0.027	0.028
	56	1.21	1.85	1.17	2.42	0.328	0.528	0.335	0.579
*Selenomonas_3*	14	0.38 ^b^	0.49 ^b^	0.42 ^b^	0.85 ^a^	0.054	<0.001	0.002	0.001
	56	1.04	1.87	1.07	1.58	0.198	0.403	0.660	0.845

^a,b^ within a row followed by different lower-case letters indicate that values differ significantly from each other (*p* < 0.05). ^1^ A_0_ contains 0 g AMT/kg DMI (DM basis); A_20_ contains 20 g AMT/kg DMI (DM basis); A_50_ contains 50 g AMT/kg (DM basis); A_80_ contains 80 g AMT/kg DMI (DM basis). ^2^ T = Treatment; L = Linear of effect of AMT; Q = Quadratic effect of AMT.

**Table 5 animals-11-03236-t005:** The effect of supplementary *Astralagus membranaceus* (AMT) on antioxidant indices in meat tissue and small intestinal mucosa of Tibetan sheep.

Items ^1^	Treatment ^2^				SEM	*p-*Value ^3^		
	A_0_	A_20_	A_50_	A_80_		T	L	Q
Meat								
CAT (U·mg^−1^)	6.43 ^b^	6.05 ^b^	7.84 ^a^	6.85 ^ab^	0.239	0.031	0.429	0.365
T-AOC (U·mg^−1^)	0.91	1.30	1.20	1.13	0.061	0.162	0.014	0.051
SOD (U·mg^−1^)	8.7 ^b^	10.1 ^ab^	10.8 ^a^	10.9 ^a^	0.278	0.033	0.337	0.624
MDA (nmol·mg^−1^)	0.65	0.67	0.62	0.63	0.011	0.401	0.914	0.620
Small intestinal mucosa								
CAT (U·mg^−1^)	5.75	7.53	7.36	8.21	0.381	0.703	0.353	0.310
T-AOC (U·mg^−1^)	2.49 ^b^	2.34 ^b^	2.82 ^a^	2.58 ^ab^	0.062	0.041	0.617	0.689
SOD (U·mg^−1^)	10.2 ^b^	16.0 ^a^	15.5 ^a^	19.4 ^a^	1.066	0.002	0.097	0.004
MDA (nmol·mg^−1^)	0.74	0.81	0.69	0.69	0.021	0.199	0.230	0.126
sIgA (μg/mg)	0.37 ^b^	0.53 ^b^	1.14 ^a^	1.14 ^a^	0.110	0.001	0.164	0.395

^a,b^ within a row followed by different lower-case letters indicate that values differ significantly from each other (*p* < 0.05). ^1^ CAT = catalase; T-AOC = total antioxidant capacity; SOD = superoxide dismutase; MAD = malonic dialdehyde; sIgA = secretory immunoglobulin A. ^2^ A_0_ contains 0 g AMT/kg DMI (DM basis); A_20_ contains 20 g AMT/kg DMI (DM basis); A_50_ contains 50 g AMT/kg DMI (DM basis); A_80_ contains 80 g AMT/kg DMI (DM basis). ^3^ T = Treatment; L = Linear of effect of AMT; Q = Quadratic effect of AMT.

## Data Availability

The rumen microbial sequencing datasets can be found in online repositories: NCBI SRA BioProject (accession no. PRJNA761911). The other data presented in this study is available upon request from the corresponding author.

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
