# Peer review of "Astragalus membranaceus Alters Rumen Bacteria to Enhance Fiber Digestion, Improves Antioxidant Capacity and Immunity Indices of Small Intestinal Mucosa, and Enhances Liver Metabolites for Energy Synthesis in Tibetan Sheep"

_animals, 2021, doi:10.3390/ani11113236_

Round 1
Reviewer 1 Report
The purpose of this study was to evaluate the effects of ingestion of 3 doses of herb Astragalus membranaceus (0, 20, 50 and 80 g/kg diet) to Tibetan sheep fed with concentrate and oat hay on rumen bacteria composition, liver metabolome response, and meat and intestinal mucosa antioxidant capacities and immunity indices. Thus, the subject falls within the general scope of the journal.
This manuscript reports a topic pertinent to contemporary. The manuscript has novelty knowledge and is well written and organized; however, there are few little flaws which should be rectified before publication.
Abstract
L34: remove “intestinal mucosa of Tibetan sheep”
Table 1. Table 1. NDF in concentrate was not detected?? Very strange!!! According to the concentrate composition the NDF concentration will be no less than 4-5%
Mat and Methods
L81: was offered as-feed basis?
L91: was collect via oral
L92: oral stomach tubing?
Please, describe the diet (DM, NDF, ADF) and AMT (Flavonoids and Astragaloside) analyses
Statistical analyses: Given the nature of the treatments (levels), it would have been convenient to analyze them as orthogonal polynomials
Results
L155: Describe “OTUs” before abbreviation
Tables 3 and 4. Please describe the units. i.e. rumen bacteria composition (as percentage of?? As log10 copies/mL? as..?)
L292: delete “Tibetan sheep” is left over
L305: which indicate “the potential” contribution of AMT to…
L332: : delete “Tibetan sheep” is left over
Reviewer 2 Report
In this paper, the authors examined the supplementation effect of well-known medical herb Astragalus membranaceus root (AMT) on rumen microbiome, antioxidant capacities, immunity indices and liver metabolome responses. The results obtained in this study could help improve sheep farming.
Minor points
Line 22: Antibiotics are natural origin. They are compounds produced by bacteria and fungi, therefore they are not chemical products. Then, please remove “in particular”, add the word “and” as “…. drugs and antibiotics.”
Line 24: Please remove word „chemical” from sequence “chemical antibiotics” or replace the word “antibiotics” with “medications” from this sentence. It will be good if you use the formula: “chemical medications”.
Line 155 please explain abbreviation OTUs.
Line 156. It is unclear what means Chao1 index? Please explain this index meaning.
Line 157. What means Shannon index? Please explain this index meaning.
Line 159. It is unclear what means Simpson index? Please explain this index meaning.
Line 208 there is A8 instead A80.
